# Next-Generation Endoscopy in Inflammatory Bowel Disease

**DOI:** 10.3390/diagnostics13152547

**Published:** 2023-07-31

**Authors:** Irene Zammarchi, Giovanni Santacroce, Marietta Iacucci

**Affiliations:** APC Microbiome Ireland, College of Medicine and Health, University College Cork, T12 R229 Cork, Ireland; izammarchi@ucc.ie (I.Z.); gsantacroce@ucc.ie (G.S.)

**Keywords:** artificial intelligence, chromoendoscopy, confocal laser endomicroscopy, dysplasia, endocytoscopy, inflammation, molecular endoscopy

## Abstract

Endoscopic healing is recognized as a primary treatment goal in Inflammatory Bowel Disease (IBD). However, endoscopic remission may not reflect histological remission, which is crucial to achieving favorable long-term outcomes. The development of new advanced techniques has revolutionized the field of IBD assessment and management. These tools can accurately assess vascular and mucosal features, drawing endoscopy closer to histology. Moreover, they can enhance the detection and characterization of IBD-related dysplasia. Given the persistent challenge of interobserver variability, a more standardized approach to endoscopy is warranted, and the integration of artificial intelligence (AI) holds promise for addressing this limitation. Additionally, although molecular endoscopy is still in its infancy, it is a promising tool to forecast response to therapy. This review provides an overview of advanced endoscopic techniques, including dye-based and dye-less chromoendoscopy, and in vivo histological examinations with probe-based confocal laser endomicroscopy and endocytoscopy. The remarkable contribution of these tools to IBD management, especially when integrated with AI, is discussed. Specific attention is given to their role in improving disease assessment, detection, and characterization of IBD-associated lesions, and predicting disease-related outcomes.

## 1. Next-Generation Endoscopy in IBD

The field of endoscopy has witnessed remarkable advancements, driven by continuous improvements in several interconnected elements that lead to overall enhancement in image quality and interpretation [1]. This is crucial in IBD for the assessment of mucosal inflammation, early detection, and precise characterization of IBD-associated lesions.

The achievement of endoscopic remission is considered a primary long-term treatment goal in Inflammatory Bowel Disease (IBD) [2]. Endoscopic mucosal healing, characterized by the resolution of visible mucosal and histological inflammation, has been associated with sustained clinical remission, reduced rates of hospitalization, and colectomy [3,4,5,6]. However, it is essential to note that there can be discrepancies between endoscopic and histologic findings. Endoscopic mucosal remission may not necessarily reflect histologic mucosal healing, especially when using white-light standard endoscopy [7]. A study by Bryant et al. revealed that 24% of ulcerative colitis (UC) patients in endoscopic remission still exhibited persistent inflammation [8]. A recent systematic review and meta-analysis [9] showed that, among patients in endoscopic remission, those who achieved histological remission (HR) had a 63% lower risk of clinical relapse compared to patients with persistent histologic activity. Advanced techniques can accurately detect endoscopic features of inflammation, bringing endoscopy closer to histology, which is crucial for effective therapeutic management.

Patients with long-standing UC and colonic Crohn’s disease (CD) have a 2-fold increased risk of developing CRC [10]. However, detecting dysplasia in the context of IBD is particularly challenging due to inflammation, and dysplastic lesions associated with IBD tend to be more frequently subtle and flat [11]. Advanced techniques are crucial for both detection and assessment of IBD-associated lesions, aiding in predicting histology and providing valuable guidance for potential endoscopic resection and a surgery-sparing approach.

Advances in endoscopic techniques include the design of endoscopes, advancement in light sources, enhanced image resolution, and improved processing and interpretation of acquired images. Notably, the evolution from standard white-light endoscopy (SD-WLE) to more sophisticated technologies has revolutionized the performance of endoscopic equipment [1]. Moreover, improvements in image processing have facilitated enhanced macroscopic evaluations of colonic mucosa through techniques such as chromoendoscopy, involving dye-based chromoendoscopy (DCE) or virtual electronic chromoendoscopy (VCE). In addition, the development of endoscopic tools able to magnify images up to 1400-fold, such as endocytoscope and probe-based confocal laser endomicroscopy (pCLE), has made in vivo histological examinations a tangible goal [12].

This narrative review aims to provide an overview of the latest advanced endoscopic techniques and their contribution to managing IBD. Specifically, the review discusses the potential role of these tools in improving disease assessment, enhancing the detection and characterization of IBD-associated lesions, and predicting disease-related outcomes.

## 2. Dye-Based Chromoendoscopy and Dye-Less Chromoendoscopy

DCE is a technique used to enhance the visualization of colorectal mucosa by employing absorbed (methylene blue 0.1%) or non-absorbed dye (indigo carmine 0.1–0.5%) at lower or higher concentrations, whether the aim is lesion detection or characterization, respectively.

As a result, the contrast between abnormal lesions and the surrounding normal colorectal mucosa is heightened, allowing for more accurate detection and characterization of the mucosal surface and borders of colonic lesions. It enhances the ability to predict histology accurately and guide appropriate therapy [13].

VCE can be obtained through different methods, including optical filters in the light source by narrowing the bandwidth of spectral transmittance (NBI) or digital post-processing for computed spectral estimation for enhancing tissue contrast (i-SCAN or Fujifilm technologies) [14].

NBI technology, pioneered by Olympus (Tokyo, Japan), employs red–green–blue filters to modify WLE. This results in improved definition of mucosal surface structures and increased contrast between superficial and deep blood vessels [15]. Furthermore, the introduction of a novel dual-focus capability (Olympus Co., Tokyo, Japan) enables optical magnification of up to 65× in near focus (NF) compared with 52× in standard focus (SF) [16]. The new EVIS X1 system has two other technologies: Texture and Color Enhancement Imaging (TXI) and Red Dichromatic Imaging (RDI). TXI enhances image color, structure, and brightness to provide a clearer definition of subtle tissue differences [17]. RDI utilizes an additional amber LED strongly absorbed by deep blood vessels, making them appear darker and, therefore, more visible [18].

The iSCAN technology developed by Pentax (Tokyo, Japan) is based on surface enhancement, tone enhancement, and contrast enhancement. The recent i-SCAN optical enhancement (OE) function, through the combination of filters producing bandwidth-limiting light and image enhancement processing technology, improves mucosal characterization and detection of lesions [19]. Furthermore, the recently introduced INSPIRA processor maintains i-SCAN digital enhancement and provides detailed images with high resolutions and high contrast [20].

Finally, Fujifilm technology (Tokyo, Japan) encompasses two components: Blue-Light Imaging (BLI) and Linked Color Imaging (LCI). BLI emits direct blue light with a short wavelength, which is selectively absorbed by haemoglobin, resulting in bright and high-contrast imaging [21]. LCI enhances red and white areas, making red areas appear redder and white areas appear brighter [22].

### 2.1. Chromoendoscopy for Disease Assessment and Outcome Prediction

There are few studies on the role of DCE to assess disease inflammation [23], predicting disease extent and histological activity [24] in IBD, since its primary application lies in detecting dysplasia in patients with long-standing colitis. On the contrary, VCE has demonstrated high accuracy in identifying mucosal inflammation, with a strong correlation with histology.

A systematic review and metanalysis [7] of 12 studies found no significant differences between the pooled correlation coefficients of endoscopic and histological scores in SD-WLE (ρ = 0.74), HD-WLE (ρ = 0.65). and VCE (ρ = 0.7). However, on 4 studies evaluating the accuracy of diagnosing HR, VCE was significantly superior to WLE (RR 1.13; 95% CI 1.07–1.19, *p* < 0.001). Similarly, a prospective cohort study by Iacucci et al. showed a significant correlation and high accuracy with histology in UC patients. The iSCAN OE score correlated significantly with the Extent, Chronicity, Activity, and Plus (ECAP) system (r = 0.7, 95% CI 0.52–0.81) and Robarts histopathology index (RHI) (r = 0.61). The accuracy of iSCAN OE in detecting abnormalities was 80% against ECAP and 68% against RHI. It was also able to identify mucosal abnormalities in quiescent patients [19] and detect abnormal vascular patterns correlated with subtle histological abnormalities [25]. Also, NBI technology has shown superiority over WLE in characterizing mucosal–vascular patterns (MVP) and predicting histology. In a prospective study on 30 quiescent/mildly active UC patients, NBI allowed a more intense visualization of vessel structures, enabling the distinction between clear and obscure MVP [26]. Biopsies from areas with obscure MVP showed significantly increased inflammatory cell infiltration (26% vs. 0%, *p* = 0.0001) and goblet cell depletion (32% vs. 5%, *p =* 0.0006) compared to areas with distorted and clear MVP [27]. Increased MVP intensity at NBI was also associated with significantly increased mucosal angiogenesis [28]. The distribution pattern of blood vessels assessed through NBI can also be helpful for the prediction of disease course. In one study on 52 UC patients in endoscopic remission, specific shapes of blood vessels were associated with high pathological activity (81%) or increased risk of inflammatory activity at one-year follow-up (OR 14.2, 95% CI 3.3–60.9) [29].

TXI and RDI are new recently introduced technologies. Thus, only some promising data about their use in IBD are present. In a prospective observational study, 146 UC patients in endoscopic remission were evaluated with WLE and TXI. Patients with accentuated redness and poor visibility of deep vessels at TXI had significantly lower UC relapse-free rates than patients with no redness or accentuated redness alone, suggesting a possible role of TXI in guiding treatment intensification [30]. Another study evaluated the correlation between endoscopic scores developed with RDI and WLI and histology in UC. RDI score was more strongly correlated with NHI (r = 0.63) than UCEIS (r = 0.51) and MES (r = 0.48) [31].

The ability to evaluate capillary patterns and the improved identification of features typical of mild inflammation made the development of a new endoscopic score based on VCE possible. The Paddington International Virtual Chromoendoscopy Score (PICaSSO) was developed and validated for UC on the iSCAN platform, demonstrating an accuracy of 72% (95% CI 64–79%) in assessing histological inflammation using RHI and 83% (95% CI 76–88%) using ECAP [32]. The score was then reproduced on NBI and FICE platforms, confirming high accuracy and excellent interobserver agreement (ICC 0.825) [33]. In the prospective multicenter study involving 307 patients, the correlation between PICaSSO and histology was strong and significantly superior to the one of Mayo endoscopic subscore (MES) and ulcerative colitis endoscopic index of severity (UCEIS). A PICaSSO score ≤3 detected HR with AUROC 0.90 (95% CI 0.86–0.94) and 0.82 (95% CI 0.77–0.87) by RHI and the Nancy Histological Index (NHI), respectively. Furthermore, it predicted better outcomes than PICaSSO > 3 (hazard ratio 0.19 [0.11–0.33] and 0.22 [0.13–0.34] at 6 and 12 months follow up, respectively) and similar to HR [34].

### 2.2. Chromoendoscopy for Colonic Lesion Detection

Guidelines [35,36] and consensus [37] recommend using DCE or VCE with targeted biopsies instead of random biopsies in IBD surveillance.

Various studies have compared DCE with SD-WLE and HD-WLE [38,39]. A randomized trial by Iacucci et al. found that HD-WLE alone was sufficient for detecting dysplasia, adenocarcinoma, or all neoplastic lesions [40]. Similarly, two retrospective studies observed no difference in dysplasia detection rate when comparing DCE with HD-WLE (10.2% vs. 6.7%, *p* 0.39) [41] or with WLE with targeted and random biopsies (11% vs. 10%, *p* = 0.8) [42], respectively. Nonetheless, other studies have emphasized the superiority of DCE over WLE in lesion detection. A first randomized prospective study by Kiesslich et al. involving 174 UC patients compared DCE and WLE. A higher number of intraepithelial neoplasia was detected using DCE (32 vs. 10, *p* = 0.003) [24]. Similarly, a metanalysis of 8 trials showed that DCE outperformed WLE for the number of dysplastic lesions detected (RR 1.8; 95% CI 1.2–2.6—absolute risk increase 6%, 95% CI 3–9%) [37]. Interestingly, a retrospective study on 110 IBD patients undergoing colonoscopy for surveillance revealed that DCE was superior to WLE in detecting nonpolypoid dysplasia [risk difference 11.8%, 95% CI (0.9–22.6), *p* = 0.008], but not for polypoid lesions (*p* = 0.12) [43].

Despite the good diagnostic performance of DCE, one drawback is the longer withdrawal time compared to HD-WLE and VCE [37,41,44], making its use challenging in a specific clinical setting. Efforts have been made to overcome this disadvantage, such as the development of a per-oral methylene blue formulation (MB-MMX) that delivers the stain directly to the colon. A randomized phase 3 trial involving 1249 patients undergoing colonoscopy for screening or surveillance demonstrated the superiority of MB-MMX over placebo in terms of adenoma detection rate (56.29% vs. 47.81%; OR 1.46), detection of nonpolypoid lesions (43.92% vs. 35.07%; OR 1.66) and adenomas ≤ 5 mm (37.11% vs. 30.9%, OR 1.36) [45]. However, further studies are needed to validate its efficacy, especially in IBD patients where active inflammation and/or the presence of pseudopolyps [46] reduce the efficacy of DCE.

Regarding VCE, several studies have compared its diagnostic performance against WLE for dysplasia detection in IBD [47,48]. In the multicentric VIRTUOSO trial involving 188 IBD patients, no significant differences were found between VCE and HD-WLE (14.9% vs. 24.2%, *p* = 0.14), with similar withdrawal times [49]. Another prospective study that evaluated 159 long-standing UC patients using HD-WLE and NBI [50] found similar rates of intraepithelial neoplasia detection and shorter withdrawal time for VCE, since fewer biopsies were required (13 min vs. 23 min, *p* < 0.001). Moreover, comparisons of different VCE platforms (i-SCAN [51], FICE [52], and NBI [53]) with DCE showed no significant differences in dysplasia detection rate or diagnostic accuracy.

In summary, VCE should be the preferred method for routine dysplasia detection, while the use of DCE should be reserved for specific patient populations at higher risk of dysplasia, such as those with primary sclerosing cholangitis or a previous history of dysplasia [24,35].

### 2.3. Chromoendoscopy for Colonic Lesion Characterization

DCE enhances dysplasia characterization compared to WLE by improving the definition of mucosal surface and lesion borders. Since endoscopic appearance can already provide an approximate assessment of the depth of infiltration, a better assessment of lesions with this technique according to the Paris classification can help in guiding endotherapy [54,55].

However, similarly to the development of endoscopic scores for grading disease inflammation, new classifications have been created to characterize endoscopic lesions better using VCE platforms. They offer more precise guidance for therapeutic management and help distinguish neoplastic from nonneoplastic lesions.

The NBI International Colorectal Endoscopic (NICE) [56] and the Japan NBI Expert Team (JNET) classification [57] were developed using the NBI platform without and with magnification, respectively. They both focus on vascular and surface patterns to predict histology and the eventual presence and depth of submucosal invasion, guiding therapeutic management.

However, certain features traditionally used for lesion characterization, such as the Kudo pit pattern, may need to be more reliable in active IBD. Therefore, the Frankfurt Advanced Chromoendoscopic IBD Lesions (FACILE) classification considering morphology, surface characteristics, vessel architecture, and lesion borders has been developed specifically for IBD-associated lesions. The presence of nonpolypoid lesions (OR 3.13), irregular vessel architecture (OR 3.49), signs of inflammation within the lesion (OR 2.42), and irregular surface pattern (OR 8.89) were identified as predictors of neoplasia [58].

To conclude, DCE and VCE are valuable instruments in the endoscopic armamentarium for grading IBD disease activity, achieving superior visualization of MH compared to conventional endoscopy, and predicting outcomes. These techniques are widely recognized since they play a crucial role in detecting and characterizing dysplasia, optimizing endotherapy according to guidelines and local expertise.

## 3. Probe-Based Confocal Laser Endomicroscopy

Probe-based confocal laser endomicroscopy (pCLE) was introduced more than 15 years ago. It is performed with a probe passed through the accessory channel of the endoscope in combination with the intravenous administration of fluorescein. The laser penetrates the mucosa at a depth of up to 250 μm, enabling an optical biopsy for real-time histological diagnosis. It allows the visualization of wider areas than conventional histology, with a more precise guiding of targeted sampling.

pCLE can provide information on structural and functional changes in the mucosa of IBD patients. By assessing crypt structure, microvessels, cells shedding, epithelial gaps, and intraluminal and intracryptic fluorescein leakage, pCLE provides a comprehensive structural and functional assessment of the intestinal epithelium. This allows clinicians to gain insights into the dynamic changes and functional abnormalities of the mucosa in IBD patients, going beyond the static information provided by conventional histology [59]. pCLE is, therefore, considered to be a reliable technique for in vivo, real-time assessment of intestinal barrier function, inflammatory activity, and detection of colonic lesions in IBD patients.

### 3.1. pCLE for Disease Assessment and Prediction of Outcome

By enabling real-time visualisation of the intestinal barrier, pCLE provides the opportunity to identify early signs of barrier disfunction and assess the efficacy of therapeutic intervention to restore barrier integrity. Incorporating these innovative imaging techniques into routine clinical practice can enhance our understanding of IBD pathogenesis, guide treatment decisions, and improve patient outcomes.

Studies evaluating 27 patients with CD [60] and 22 with UC [61] found the decreased number and increased tortuosity of crypts, dilated crypt lumen, microerosions, increased cellular infiltrates in the lamina propria, increased vascularization, and decreased number of goblet cells as features of active disease. Moreover, even quiescent CD [60] and UC [62] patients may have chronic changes, especially when compared to healthy subjects.

Notably, studies have demonstrated that achieving remission as defined by pCLE is more closely associated with HR than WLE. In a prospective study on 73 UC patients [63], more than half of the patients in WLE-defined remission were found to have active inflammation at histology, while those in remission, according to pCLE, were also in HR. Crypt architecture and microvascular alterations assessed by CLE strongly correlated with the Geboes histological score (ρ 0.738 and 0.617, respectively).

Neumann et al. proposed a score that considers the number of goblet cells and crypts, enabling the prediction of HR in apparently noninflamed areas at endoscopy [64]. Similarly, Hundorfean in a pilot study on 10 UC patients created a score that strongly correlated with histology (r = 0.82 by Gupta score) and showed a high accuracy (94.44%) for mucosal healing. Patients were evaluated before and after treatment, showing a consensual modification in the score assessed by CLE [65].

Importantly, endomicroscopy features correlating with histology have also demonstrated the ability to predict outcomes. Kiesslich et al. showed that alteration of intestinal barrier function by increased cell shedding and fluorescein leakage was associated with relapse after 12 months [66] in IBD patients in clinical remission. Similarly, another work found that signs of active inflammation at pCLE could predict relapse with an accuracy of 74.4% in UC patients in clinical remission [67]. Furthermore, pCLE has shown the potential to predict postoperative recurrence in patients with CD [68].

Recent research studies have highlighted the role of the intestinal barrier in the IBD natural history, demonstrating a correlation between barrier dysfunction, clinical disease behaviour, and long-term disease outcomes [66,69]. Rath et al. evaluated 181 IBD patients in clinical remission. pCLE-assessed barrier healing was superior to endoscopic and HR in predicting survival free from major adverse outcomes (MAO). In particular, 19.1% of UC and 29.6% of CD patients with barrier healing experienced MAO, compared with 44.4% and 51% among the ones who achieved combined endoscopic and HR [70], respectively. Moreover, in a study by Iacucci et al., vessel tortuosity, crypt morphology, and fluorescein leakage predicted response to treatment in UC (AUROC 0.93; accuracy 85%) and CD (AUROC 0.79, accuracy 80%) [71].

### 3.2. pCLE for Colonic Lesions Detection and Characterization

A meta-analysis of 9 studies reported a pooled sensitivity of 87%, specificity of 94%, and AUROC 0.96 for differentiating neoplastic lesions from nonneoplastic ones using pCLE [72]. Additionally, the utility of pCLE in the surveillance of high-risk patients was investigated in a trial on 69 PSC-IBD patients and demonstrated an accuracy of 96% and an NPV of 99% in differentiating neoplastic from nonneoplastic mucosa [73].

Furthermore, pCLE and conventional colonoscopy were compared in a randomized prospective study involving 161 UC patients in clinical remission. An almost 5-fold increased neoplasia detection rate was found for CLE, with an accuracy in predicting neoplastic changes of 97.8% [74]. When assessing dysplasia detection, pCLE exhibited a sensitivity of 100% and specificity of 90% compared to histology [75]. Unfortunately, pCLE assessment remains operator-dependent and requires prior training. Concerns remain about the cost of technology and timing of procedures required.

In summary, pCLE is an innovative endoscopic technique that offers extraordinary capabilities in assessing barrier structure and function. While currently primarily used in research settings, pCLE holds great promise in clinical practice for evaluating mucosal inflammation, predicting outcomes, and studying colitis-related colonic lesions.

## 4. Endocytoscope

The endocytoscope is an innovative ultra-high magnification technique enabling an enhancement of up to 450 to 1400 times, providing detailed visualization of cellular structures. It requires the application of a mucolytic agent, such as N-acetylcysteine, to facilitate the penetration of a topical contrast agent, which can be methylene blue, toluidine blue, or cresyl violet [76].

### 4.1. Endocytoscope for Disease Assessment and Outcome Prediction

The endocytoscope has proven to be a valuable tool for assessing the ultrastructure of the intestinal barrier. It enables the visualization of crypts architecture, infiltration of cells and microvessels for IBD disease assessment, and colitis-related dysplasia [77].

In a retrospective study by Maeda et al. on UC patients, combined evaluation with NBI and endocytoscope was superior to conventional endoscopy for specificity, NPV, and accuracy (100%, 100%, and 92.3%, respectively) in detecting acute inflammation [78]. Similarly, another study revealed a strong agreement between the disease activity assessed by endocytoscope and the Geboes score (ICC 0.78 (95% CI 0.67–0.86)), with a sensitivity of 0.77, specificity 0.97 (95% CI 0.83–0.99), and a diagnostic accuracy of 0.86 (95% CI 0.75–0.93) for HR [79].

Several endocytoscopic scoring systems (ECSS) in IBD have been developed to grade disease activity accurately. Vitali et al. developed a score overcoming WLE for the detection of microscopic activity, with a sensitivity of 88%, specificity of 95.2%, AUROC 0.916, and a strong correlation with RHI (0.7) and NHI (0.73). Disease remission defined according to this ECSS score was shown to predict adverse outcomes with similar accuracy to histology [80]. Likewise, another endocytoscopic score taking into account crypt architecture, the distance between crypts, the infiltration of cells among crypts, and microvessels, showed a strong correlation with endoscopic and histological scores (RHI r = 0.89 and NHI r = 0.86) [81,82]. Additionally, in a prospective study on 32 UC patients, 4 groups were identified according to the crypt and pit architecture. Altered endocytoscopic features were associated with disease relapse, while none of patients with a normal appearance at endocytoscopy experienced a flare-up [83]. Similarly, Maeda et al. found that the risk of clinical relapse could be stratified according to intramucosal capillary network and crypt architecture evaluated by endocytoscope [84].

### 4.2. Endocytoscope for Colonic Lesions Detection and Characterization

The available data on the diagnostic performance of the endocytoscope in IBD surveillance are currently limited, and its potential role in the assessment of dysplasia in IBD has been hypothesized [85]. In a prospective study on 60 non-IBD patients, the endocytoscope demonstrated an ability comparable to histology to distinguish neoplastic and nonneoplastic colorectal lesions, with an overall accuracy of 93.3% and very strong agreement with histology (k 0.91) [86].

Further investigations are necessary to assess the sensitivity, specificity, and accuracy of endocytoscope in detecting dysplasia and its potential as a surveillance tool in IBD. These studies will provide valuable insight into the diagnostic capabilities and practical applicability of the endocytoscope, ultimately contributing to its integration into routine clinical practice for IBD patients (Figure 1).

In conclusion, the endocytoscope, by offering real-time cellular imaging, holds great promise in providing valuable information on inflammation and dysplasia and could play a crucial role in guiding IBD management and endotherapy in the future.

However, its use is still limited to research and in tertiary centers.

## 5. Future Generation of Imaging in IBD

### 5.1. Image Interpretation beyond the Human Eye—Artificial Intelligence

The interpretation of medical images is still challenging since it is characterized by high interobserver variability and low agreement, even among experts. This variability is particularly evident as imaging technologies become more complex, less widespread, and require extensive training.

Artificial intelligence (AI) represents the ability of machines to imitate human intelligence (Figure 2).

It has the potential to detect subtle elements that may be missed by the human eye and assist in the characterization of identified lesions. By employing specific algorithms, computers can utilize existing data to develop appropriate models. These machine learning (ML) models can then be applied to evaluate new clinical scenarios. As a subset of ML, deep learning with a convolutional neural network (CNN) is fully automated and a fast process that can accurately analyze images [87].

#### 5.1.1. AI for Disease Assessment and Outcome Prediction

Various studies have developed and utilized deep learning models and convolutional neural networks (CNN) to assess endoscopic activity and provide comparisons with expert opinions in UC.

A CNN algorithm demonstrated excellent performance in distinguishing endoscopic remission from moderate-to-severe disease, with an AUROC of 0.966 (95% CI, 0.967–0.972), a sensitivity of 83.0% (95% CI, 80.8–85.4%), and specificity of 96.0% (95% CI, 95.1–97.1%), respectively [88]. Additionally, another CNN exhibited good accuracy for MES and UCEIS, with a very good agreement (k 0.8) with endoscopist scores [89].

In a multicenter cross-sectional study, Takenaka et al. developed a deep neural network for UC (DNUC) to evaluate histological and endoscopic activity by UCEIS in patients in clinical remission (180) and activity (590), respectively, and assess the consistency with experts. The system exhibited a sensitivity of 97.9% and a specificity of 94.6% for predicting HR. The agreement for endoscopic scoring by UCEIS between DNUC and experts was excellent (ICC of 0.927) [90]. Furthermore, the DNUC system could also predict clinical outcomes at 12 months [91].

Similarly, Iacucci et al. developed the first AI-based system to assess endoscopic activity and predict the histological activity and clinical outcomes on WLE and VCE videos. Starting from 1.090 videos of the multicentric PICaSSO study, a CNN able to detect endoscopic remission was built. The algorithm showed a sensitivity, specificity, and AUROC of 72%, 87%, and 0.85 for WLE, while for VCE all parameters improved (79%, 95%, and 0.94, respectively). The same algorithm could also predict HR and specified clinical outcomes in 12 months [92].

In a pilot study on 29 UC patients, an algorithm based on the pixel color, the red channel of the RGB pixel values, and vascular pattern recognition was able to estimate disease activity in UC (Red Density, Pentax) [93]. This objective operator-independent score significantly correlated with MES, UCEIS, and RHI (r 0.76, 0.74, and 0.74, respectively).

Maeda et al. developed a computer-aided diagnosis system based on endocytoscopy. The system exhibited a diagnostic accuracy of 91% (95% CI 83–95%) for predicting HR [94]. Subsequently, patients were classified into AI-healing or AI-histologically active groups. The model predicted clinical relapse at 12 months, with a higher rate observed in the AI-histologically active group (28.4 vs. 4.9%, *p* < 0.001).

Recently, Gui et al. evaluated, for the first time, the applicability of a novel histological score for UC, named the PICaSSO histologic remission index (PHRI), in an artificial intelligence system. Starting from 614 biopsies of 307 UC patients with UC, a CNN was created to distinguish active and quiescent UC using a subset of 138 biopsies. The algorithm showed a sensitivity of 78%, specificity of 91.7%, and accuracy of 86% [95] for the presence or absence of neutrophils. Subsequently, another CNN was trained from a subset of 118 biopsies to distinguish HR from the activity and predict the flare-up at one year. Sensitivity for histological activity/remission was 89%, 94%, and 89% for PHRI, RHI, and NHI, while specificity was 85%, 94%, and 76%, respectively. AI-assessed PHRI also allowed a better risk stratification for disease relapse at one year (4.64 vs. 3.56) [96].

#### 5.1.2. AI for Lesion Detection and Characterization

Computer-aided systems able to detect dysplastic lesions and predict histology have been developed, but mainly in non-IBD populations. Few systems have been assessed on IBD patients [97,98].

Starting from computer-aided detection systems previously developed for non-IBD patients, one system was retrained on IBD-associated lesions, including pseudopolyps and serrated epithelial changes. The system showed a high sensitivity (95%), specificity (98.8%), and accuracy (96.8%) on HD-WLE in detecting IBD dysplastic lesions, as well as pseudopolyps, without being significantly influenced by inflammation severity [98].

Moreover, Kudo et al. [99] showed that an AI-assisted system (EndoBRAIN^®^) developed on 69.142 endocytoscopic images was reliable in differentiating neoplastic from nonneoplastic lesions in non-IBD patients. On analysis with NBI, EndoBRAIN^®^ distinguished neoplastic from nonneoplastic lesions with all diagnostic performance parameters significantly higher than trainees, and sensitivity and NVP higher than experts: 96.9% sensitivity (95% CI, 95.8–97.8), 94.3% specificity (95% CI, 92.3–95.9), 96.0% accuracy (95% CI, 95.1–96.8).

#### 5.1.3. Limitations of AI Applications

Some limitations of artificial intelligence may affect its reliability, also in endoscopy, and therefore standardization and guidelines are required.

One significant challenge is the dependence on sample size, as AI models require large and various datasets to be trained effectively. However, obtaining such datasets can be difficult due to privacy and data protection concerns [100] and the limited availability of annotated endoscopy images and videos [101]. Furthermore, data security is a significant concern, given the role entities owning AI technologies play in obtaining, utilizing, and protecting patient health information. Additionally, there is the risk of reidentifying previously anonymized patients through AI-driven methods. Therefore, it is crucial to implement robust cybersecurity strategies at all stages of algorithm development to ensure and uphold patient privacy [102].

Moreover, the lack of diversity in the training data can lead to biased and suboptimal AI algorithms, potentially compromising their accuracy in real-world clinical settings [103]. Another critical concern is the quality of data used to train AI-assisted endoscopy systems. These systems heavily rely on high-quality input data to produce reliable outcomes. Any inaccuracies or inconsistencies in the imaging equipment or data acquisition process can result in erroneous AI predictions [104,105]. Thus, ensuring the standardization of data collection methods and endoscope quality is essential to improve the robustness of AI models in endoscopy.

Implementing AI-based tools in the clinical setting also poses its own set of challenges. Physicians and medical professionals need to be adequately trained to use these AI systems effectively and interpret their outputs correctly. The integration of AI into existing clinical workflows demands coordination and may require substantial changes in the working practices of medical teams. Additionally, issues related to liability and medical ethics arise, as the responsibility for medical decisions made in conjunction with AI systems becomes a matter of concern [106]. Despite these limitations, the potential benefits of AI in endoscopy are substantial. AI has demonstrated the ability to assist in the early detection of abnormalities, improve diagnostic accuracy, and aid in real-time decision-making during endoscopic procedures. Collaborative efforts between AI developers, clinicians, and regulatory authorities are essential to overcome the limitations. Rigorous evaluation of AI algorithms on different datasets, continuous performance monitoring, and ongoing technology refinement will be crucial for its successful integration into clinical practice [107]. By addressing these challenges, AI can undoubtedly play a transformative role in advanced endoscopy and ultimately improve patient outcomes.

### 5.2. Raman Spectroscopy

Raman spectroscopy (RS) is a promising technique based on the scattering of inelastic light, which generates highly specific spectra for several molecules. Preliminary studies tried to identify differences in spectra that differentiate CD from UC [108,109] and inflammation from remission.

In a study by Smith et al., the ability of RS to differentiate inflammation from MH and eventual spectral changes before and after treatment has been evaluated in IBD patients. RS was able to accurately differentiate MH from inflammation with a sensitivity, specificity, and accuracy of 96.29%, 95.03%, and 95.65% in UC and 96.19%, 88%, and 91.6% in CD, respectively [110].

### 5.3. Molecular Endoscopy

Molecular endoscopy allows for applying labelled probes, such as antibodies, against specific target structures in the gastrointestinal tissue. The field of molecular imaging in IBD holds promising potential for personalized therapy.

Studies have focused on molecular targets such as tumor necrosis factor (TNF) and alpha (α)_4_ beta (β)_7_ integrin, assessing their expression through fluorescent-labelled antibodies and pCLE. These investigations have demonstrated associations between target expression, treatment response, and mucosal healing. In a phase II trial, the role of membrane-bound TNF (mTNF) binding by a fluorescent-labelled anti-TNF antibody visualized by CLE has been evaluated. The lower the number of mTNF+ cells, the lower the probability of clinical relapse (92% vs. 15%), with mucosal healing observed at follow-up endoscopy [111]. Similarly, in a pilot study on 5 CD patients, fluorescent antibodies directed against α_4_β_7_ integrin were applied topically, and then ex vivo CLE was used to estimate integrin expression. No response to vedolizumab was observed in the 3 α_4_β_7_ integrin- patients, and α4β7+ patients showed clinical response to treatment [112].

Finally, Iacucci et al. evaluated 15 CD and 14 UC patients before and after treatment with anti-TNF or anti-integrin. The binding of labelled biological agents was analyzed ex vivo. An increased pretreatment binding was associated with a response to treatment that was better for UC (AUROC 83%; accuracy 77%; PPV 89%; NPV 50%) than CD (AUROC 58%, accuracy 64%, PPV 40%, NPV 78%) [71].

To summarize, the future of endoscopy in IBD is undoubtedly shaped by the integration of AI models. AI has the potential to detect subtle changes that may not be visible to the human eye, thereby reducing interobserver variability and enhancing diagnostic accuracy of inflammation and dysplasia. Furthermore, AI can play a crucial role in guiding decision-making during endoscopic procedures, leading to more precise and targeted interventions. In addition to AI, the application of molecular techniques, such as molecular endoscopy and Raman spectroscopy, is an emerging area with the potential to provide personalized and tailored approaches for IBD patients.

## 6. Conclusions

Next-generation endoscopic techniques have demonstrated significant potential for improving IBD assessment, detection and characterization of colitis-associated lesions, as well as the ability to predict long-term outcomes.

In clinical practice, the routine use of DCE and VCE has already resulted in improved disease activity definition and more accurate lesion detection/characterization, guiding disease management and endotherapy in an organ-sparing perspective.

Furthermore, pCLE and endocytoscope offer the unique advantage of providing real-time disease assessment that closely correlates with histology and allows for the evaluation of intestinal barrier function. Although they are currently mainly used in research settings, these techniques hold significant potential to guide disease management in the future.

The development of AI systems capable of interpreting endoscopic images beyond the human eye can further improve disease management by facilitating standardized evaluation of disease activity and lesion characterization for each of the aforementioned techniques.

Additionally, while molecular endoscopy techniques are still in their infancy, they hold promise for personalized therapy approaches, potentially revolutionizing the field of IBD management.

Overall, the integration of next-generation endoscopic techniques, AI, and emerging molecular endoscopy approaches will provide a comprehensive framework for IBD management.

## Figures and Tables

**Figure 1 diagnostics-13-02547-f001:**
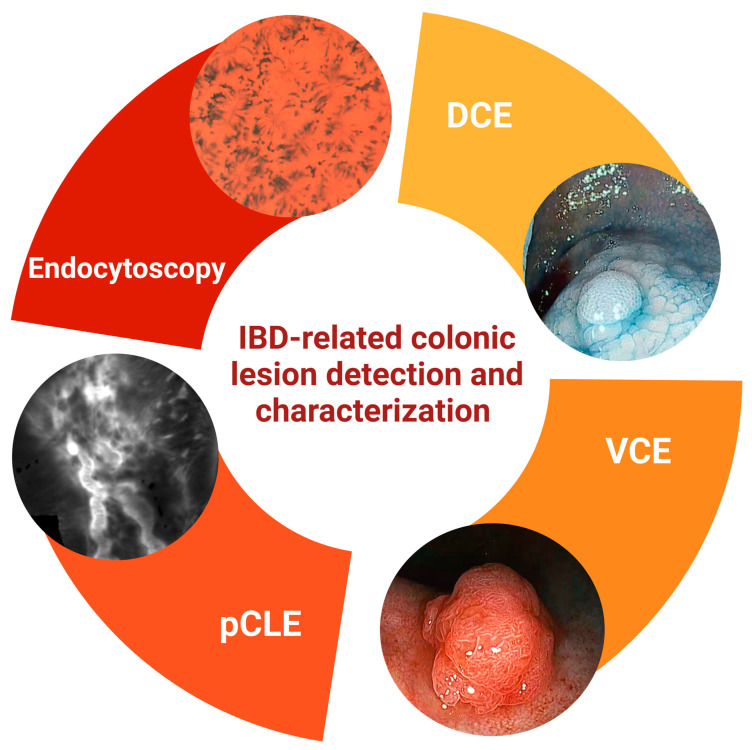
**Inflammatory Bowel Disease (IBD)-related colonic lesion detection and characterization.** In this figure is highlighted the pivotal role of advanced endoscopic techniques in the detection and characterization of colonic lesions associated with IBD. Images obtained through dye-based chromoendoscopy, virtual electronic chromoendoscopy, probe-based confocal laser endomicroscopy, and endocytoscopy are presented. Created with ‘Biorender.com’. *Abbreviations: DCE, dye-based chromoendoscopy; IBD, inflammatory bowel disease; pCLE, probe-based confocal laser endomicroscopy; VCE, virtual electronic chromoendoscopy*.

**Figure 2 diagnostics-13-02547-f002:**
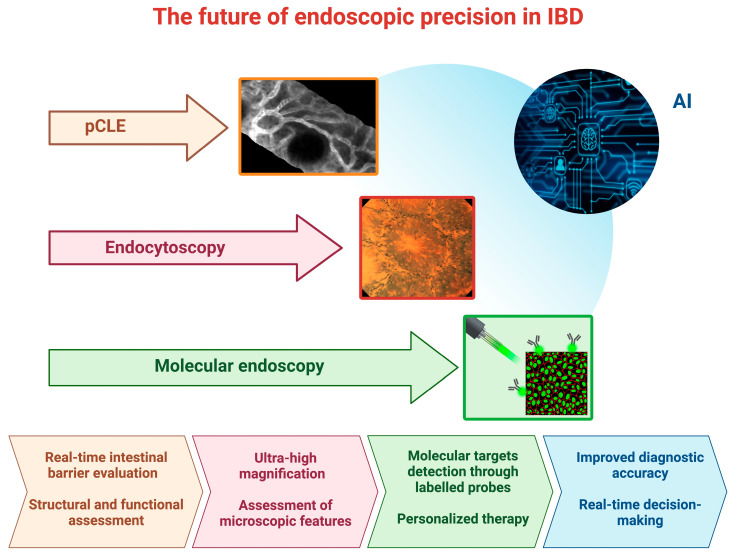
**The future of endoscopic precision in Inflammatory Bowel Disease (IBD).** This figure shows the emerging endoscopic techniques for the assessment of patients with IBD. Images captured through probe-based confocal laser endomicroscopy, endocytoscopy, and molecular endoscopy are presented. Also, the figure highlights the potential of artificial intelligence in enhancing and optimizing these techniques. The lower section of the figure provides a summary of the improvements associated with these advanced tools, highlighting their role in personalizing therapy and enabling real-time decision-making during endoscopic procedures. Created with ‘Biorender.com’. *Abbreviations: AI, artificial intelligence; IBD, inflammatory bowel disease; pCLE, probe-based confocal laser endomicroscopy*.

## Data Availability

Not applicable.

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
