# Peer review of "Next-Generation Endoscopy in Inflammatory Bowel Disease"

_diagnostics, 2023, doi:10.3390/diagnostics13152547_

Round 1

Author Response

Dear Editorial Office,

Thank you for your valuable feedback. We appreciate the reviewer's comments and have carefully addressed each point to improve the quality of our review article.

Below is the point-by-point response letter.

Reviewer 1

The idea of reviewing the latest advancements in the field of endoscopic techniques focused on better mucosal characterization and detection of potential dysplastic lesions in IBD patients is very well welcomed. Overall, the quality of the article is really appreciated. However, here are some remarks that I would suggest to take into consideration:

  1. I would recommend to structure differently the first chapter "Next generation endoscopy in IBD". You

discuss about endoscopic mucosal healing/remission, then you develop the idea of how the techniques had improved over time and finally you return and discuss about dysplasia detection. It is better to have an introduction regarding the importance of high quality imagining in IBD (including what you have mentioned about endoscopic MH and dysplasia detection) and secondly to talk about the new techniques that you have previously mentioned.

Response: Thank you for your comment. The paragraph has been restructured according to your suggestions.

  1. Line 81 - "Staining can be applied through the endoscope with a pump jet or catheter spray.". I would delete this. You should strictly focus on advanced imaging and not on any technical details that are not of interest for this topic.

Response: As suggested, the sentence has been deleted.

  1. Lines 95-99 "The new EVIS X1 system has two other technologies: Texture and Colour 95 Enhancement

Imaging (TXI) and Red Dichromatic Imaging (RDI). TXI enhances image 96 colour, structure, and brightness to provide a clearer definition of subtle tissue 97 differences [17]. RDI utilizes an additional amber LED strongly absorbed by deep blood 98 vessels, making them appear darker and, therefore, more visible [18]" If you add this information into discussion, please also bring comments and data from the literature which support the idea that these particular techniques (TXI, RDI) really add a benefit for IBD patients; if not, please do not bring it into discussion.

Response: Thank you for your comment. Data about the application of TXI and RDI in IBD have been added in 2.1 section.

  1. Lines 104-105 "OE filter comprehends Mode 1, which 104 provides sufficient light and highlights blood vessels, and Mode 2, which enhances the 105 blood vessels and the mucosa in a natural color". I do not consider these details relevant for this topic.

Response: As suggested, the sentence has been deleted.

  1. I do not consider Figure 1 really adds any benefit to the reader and I suggest you to remove it. Also, regarding Figure 2, please mention if the images are from your unit; otherwise, mention the source.

Response: As suggested, the figure 1. has been removed. We confirm that the images present in Figure 2 are from our unit.

  1. Line 238 "promoting an organ-sparing approach." - The conclusion of the paragraph is fine, but I do not consider that the idea is of "organ sparing" since the techniques simply better detect and characterize the lesions. If dysplasia is detected, the further management is purely dictated by guidelines and local expertise (follow-up, EMR, ESD, segmental colectomy, etc).

Response: Thank you for your comment. The sentence has been reformulated.

  1. Lines 370-371 "In the near future, it is expected to play a crucial role in guiding IBD management and endotherapy.". I am afraid "near future" is way too optimistic. Please reformulate since many years from now on the endocytoscope will still be limited for research purposes in a very few tertiary centers.

 Response: As suggested, the sentence has been reformulated. Page 10

Reviewer 2 Report

The authors provide an excellent narrative review on the next-generation endoscopy in Inflammatory Bowel Disease and how AI can help accelerate the technology for improving care and practice. One major part that is missing is, while AI can be a useful tool, the authors should provide the limitations in applying AI for each of the items in section 5 and how it can be addressed. Like sample size issues, fidelity of the data, diversity etc. to provide reliable outcomes from AI assisted endoscopy systems. Furthermore, implementing AI based tools in a practical clinical setting is very challenging, how can this be addressed? How can we best prepare future gastroenterologists to be adaptable to AI assisted next generation endoscopy systems? The author should provide a brief discussion addressing these issues, which will be very helpful to the readers. 

Author Response

Dear Editorial Office,

Thank you for your valuable feedback. We appreciate the reviewer's comments and have carefully addressed each point to improve the quality of our review article.

Below is the point-by-point response letter.

Reviewer 2

The authors provide an excellent narrative review on the next-generation endoscopy in Inflammatory Bowel Disease and how AI can help accelerate the technology for improving care and practice. One major part that is missing is, while AI can be a useful tool, the authors should provide the limitations in applying AI for each of the items in section 5 and how it can be addressed. Like sample size issues, fidelity of the data, diversity etc. to provide reliable outcomes from AI assisted endoscopy systems. Furthermore, implementing AI based tools in a practical clinical setting is very challenging, how can this be addressed? How can we best prepare future gastroenterologists to be adaptable to AI assisted next generation endoscopy systems? The author should provide a brief discussion addressing these issues, which will be very helpful to the readers. 

Response: We thank the reviewer for the suggestion. We agree that this information will be of help to the reader.

A paragraph addressing the main limitations of AI in endoscopy has been added page 12.
